# Enhanced Preservative Performance of Pine Wood through Nano-Xylan Treatment Assisted by High-Temperature Steam and Vacuum Impregnation

**DOI:** 10.3390/ma16113976

**Published:** 2023-05-26

**Authors:** Shutong Fan, Xun Gao, Jiuyin Pang, Guanlin Liu, Xianjun Li

**Affiliations:** 1College of Materials Science and Engineering, Central South University of Forestry and Technology, Changsha 410004, China; 2College of Architecture and Energy Engineering, Wenzhou University of Technology, Wenzhou 325006, China; 3Key Laboratory of Wooden Materials Science and Engineering of Jilin Province, Beihua University, Jilin 132013, China; 4State Grid Longjing Power Supply Company, Longjing 133400, China

**Keywords:** preservative performance, pine wood, nano-xylan, high temperature and high-pressure steam, vacuum impregnation

## Abstract

This study used environmentally friendly nano-xylan to enhance the drug loading and preservative performance (especially against white-rot fungi) of pine wood (*Pinus massoniana Lamb*), determine the best pretreatment, nano-xylan modification process, and analyze the antibacterial mechanism of nano-xylan. High-temperature, high-pressure steam pretreatment-assisted vacuum impregnation was applied to enhance the nano-xylan loading. The nano-xylan loading generally increased upon increasing the steam pressure and temperature, heat-treatment time, vacuum degree, and vacuum time. The optimal loading of 14.83% was achieved at a steam pressure and temperature of 0.8 MPa and 170 °C, heat treatment time of 50 min, vacuum degree of 0.08 MPa, and vacuum impregnation time of 50 min. Modification with nano-xylan prohibited the formation of hyphae clusters inside the wood cells. The degradation of integrity and mechanical performance were improved. Compared with the untreated sample, the mass loss rate of the sample treated with 10% nano-xylan decreased from 38 to 22%. The treatment with high-temperature, high-pressure steam significantly enhanced the crystallinity of wood.

## 1. Introduction

Compared with cementitious materials, wood-based construction materials can lead to greener and cleaner infrastructure due to their low carbon emissions and environmental friendliness. However, carbohydrates and other nutrients in wood materials can serve as food sources for wood-rotting fungi that degrade wood’s internal structure and reduce its service life. In addition, where the wood materials are placed strongly influences the mechanical decay and properties of the constituent materials, especially if they are susceptible to fluctuations of temperature and relative humidity [1]. Witomski [2] studied the mechanical properties of decayed pine wood and found that the flexural strength decreased rapidly, while the compressive strength and flexural elastic modulus showed linear changes. Bouslimi [3] showed that the decreased flexural strength of *Thujia occidentalis* due to natural brown rot was caused by the degradation of arabinose and galactose. Bari [4] compared changes in the mechanical properties and major chemical components of *Fagus sylvatica* during short-term (30 d) and long-term (120 d) white rot processes. They found that the impact toughness and grain compressive strength had the highest correlation with sugar content and cellulose content (*R*^2^ = 0.96). During wood decay, the degradation of chemical components is the underlying cause of macroscopic property changes and microstructural damage to wood. When viewed at the microscale, hemicelluloses and lignin are the weak structural chemical components of the cell walls. Therefore, preservative treatment for wood materials is necessary before their utilization [5].

Current preservative treatments mainly utilize copper, organic ammonium [6], organic preservatives [7], and other preservatives [8]. However, these treatment agents can generate pollution due to their toxicity [9]. Bio-preservatives such as polyphenols, tannins, plant extracts, and chitosan are highly effective, broad-spectrum, safe, non-toxic, easily degradable, and not highly resistant to additives [10]. Victor et al. used 10-year-old teak sapwood and *Pinus* as natural alternatives to evaluate the influence of teak extracts on the natural resistance of wood. The treatments containing ethanol extractives significantly increased the resistance against brown and white-rot fungi [11]. However, biological preservatives are restricted by their single raw material sources, low extraction efficiency, high costs, and poor penetration. These result in low additive loadings in wood materials and a low anti-corrosion efficiency, which also restricts the efficient and large-scale production and use of biological preservatives. To resolve this, this study aims to apply biological preservatives for the preservative treatment of wood materials [12].

Pine wood (*Pinus massoniana Lamb*) was selected as the representative wood material in this study. Pine wood has a high strength, beautiful texture, and easy processability [13] and can be used as a high-grade civil engineering material [14]. Pine wood is used in indoor furniture, wood structure buildings, and wetland landscape facilities and is an important biomass resource. However, in the case of buildings made of wood, the variations and fluctuations in the microclimatic variables such as temperature and relative humidity, may have a detrimental effect on the mechanical properties and preservative performance of wooden objects [15]. Singh [16] utilized single-sewage sludge produced from pine wood for the phytoremediation of polluted soil to remove heavy metals. Despite this, there have been no investigations into the use of biological preservatives for pine wood.

Xylan is an inexpensive and non-toxic natural biological preservative [17,18]. It is an abundant renewable resource and is a structural polymer of plant cell walls. One of the hemicelluloses, xylans, constitute 25–35% of the dry biomass of woody tissues of dicots and lignified tissues of monocots and account for up to 50% of some tissues of cereal grains. Xylan possesses excellent antioxidant and antimicrobial properties and may destroy the outer membrane of bacterial cells and inhibit their growth [19]. However, xylan is currently mainly used in pharmaceuticals, sewage treatment, papermaking auxiliaries, food additives, and other fields, and few scholars have conducted similar studies in wood preservative treatment. Former studies have shown that xylan–chitosan nanoscale conjugates have enhanced anti-bacterial performance [20]. Thus, nanoscale xylans may have enhanced anti-fungi performance, and their utilization will reduce environmental pollution caused by currently used preservatives. Investigations into nano-xylan are quite limited, and its influence on the anti-corrosion property of pine wood is unclear.

The efficiency of any preservative treatment strongly depends on preservative retention. However, the permeability of pine wood is quite low compared with other fast-growing trees [21]. Research has shown that high-temperature, high-pressure treatment is an effective treatment method [22,23]. Sasaki [24] indicated the treatment of softwood with high-temperature, high-pressure steam enhanced the additive loading, which is also an environmentally friendly modification protocol. Gao [25] found that high-temperature, high-pressure treatment degraded hemicelluloses in wood due to its poor heat resistance and significantly reduced the number of free hydroxyl groups on the wood surface. This further enhanced the dimensional stability [26] and corrosion resistance of wood.

Besides high-pressure, high-temperature treatment, vacuum treatment can also enhance the impregnation efficiency of preservative solutions by eliminating air in the cavities and intercellular spaces [27]. This promotes the permeation of the preservative solution with the generated pressure difference between the surface and interior of the wood [28]. Furthermore, the volume expansion of wood under vacuum will increase the cell spacing and enhance the impregnation depth. Currently, related studies to improve the additive loading of biological preservatives for the treatment of pine wood are quite limited. In this paper, pine wood was first modified using high-temperature, high-pressure steam and then submerged in a nano-xylan solution for vacuum impregnation. The additive loading was optimized by adjusting the treatment parameters. Then, the treatment effect was inspected by measuring the mechanical performance of the modified specimen after fungal infection. The corresponding mechanism analysis and their mechanical and anti-corrosion properties were conducted with Fourier-transform infrared (FTIR) spectroscopy, scanning electron microscopy (SEM), and X-ray diffraction (XRD). The aim of the research was to use the nano-xylan to improve the drug loading and anti-fungal performance of pine wood using high-temperature, high-pressure steam pretreatment-assisted vacuum impregnation; determine the best pretreatment, nano-xylan modification process; and analyze the antibacterial mechanism of nano-xylan. This study provides a theoretical basis and technical support for the efficient utilization of biomass resources.

## 2. Experimental

### 2.1. Raw Material

Corn cob powder (40–100 mesh), glucose, and brown sugar were purchased from Tianjin Yongda Chemical Reagent Co., Ltd. (Harbin, China). All the chemicals used in this study (the ethanol, sodium potassium tartrate, sodium hydroxide, phenol, sodium sulfite, and sulfuric acid) were analytical purity and provided by Tianjin Damao Chemical Reagent Factory. The wood-rotting fungi (white rot, Trametes versi-color) were purchased from the Chinese Academy of Forestry Sciences. Before using, the Pine wood (heartwood, and the age of the wood was 20 years) was dried at 40 °C for 24 h until the moisture content was 15%. The Pine wood (*Pinus massoniana Lamb*) was purchased from Yong Xv Industrial Wood Composites Factory (Harbin, China).

### 2.2. Method

Preservative treatment of pine wood with synthesized nano-xylan.

In this section, the preservative treatment of pine wood with the synthesized nano-xylan was carried out in two main steps: nano-xylan synthesis and preservative pre-treatment. The average of five values obtained from identical specimens was used as the result.

#### 2.2.1. Synthesis of the Nano-Xylan from the Corncob Powder

Crude xylan was first extracted from corncob powder. Corncob powder (50 g) and 50 mL of 8% NaOH solution were first added into a three-mouth bottle. Then, the bottle was immersed in a water bath at 80 °C. The mixture was stirred at a speed of 180 rpm for 150 min. Then, the ethanol (mass fraction of 95%) with a volume ratio of 1:3 was added to the mixture, and crude xylan was obtained via centrifuging at 8000× *g* rpm for 10 min.

Nano-xylan was then synthesized with the extracted crude xylan. Crude xylan (1 g) was added to 100 mL of 2% NaOH solution at 90 °C for 20 min. The dissolution of the xylan was promoted with ultrasonic vibration at 80 W. Then, 300 mL of ethanol was added to the mixture, and nano-xylan was obtained via freeze-drying the filtered precipitate. The average particle size of the synthesized nano-xylan in solution was 76.49 nm, as measured using a nanoparticle size analyzer (Winner801, Jinan Winner Particle Instrument Stock Co., Ltd. Jinan, China).

#### 2.2.2. Pretreatment of the Pine Wood and Nano-Xylan Impregnation

Pine wood was pretreated in a high-temperature, high-pressure steam generator (Northeast Forestry University, Harbin, China), which treated samples with saturated steam and superheated steam in the temperature and pressure ranges of 60–180 °C and 0.2–1 MPa, respectively. The samples of pine wood (20 mm × 20 mm × 10 mm) were maintained under different temperatures (120, 140, 160, and 170 °C) for 20–50 min. The corresponding steam pressures were 0.2, 0.4, 0.6, and 0.8 MPa. After that, the specimens were put into a drying oven at 105 °C to reduce the moisture content to 3–5%.

The treated specimens were placed in the nano-xylan solution with a concentration between 2 and 10% and impregnated in a vacuum chamber with different vacuum degrees (−0.02, −0.04, −0.06, −0.08 MPa) for a certain time (10–50 min). Then, the specimens were oven-dried at 80 °C for 6 h and weighed to determine the additive loading, which was optimized through the response surface method [29].

#### 2.2.3. Inoculation and Cultivation of Wood-Rotting Fungi

The incubator for the fungi was prepared with brown sugar and feeding wood. The brown sugar, cornflour, sawdust (30 mesh), and river sand were first put into a conical flask and stirred evenly with a weight ratio of 1:8:14:140. Then, three pieces of feeding wood and 100 mL of maltose solution were added to the flask at 121 °C for 20 min. After that, a small piece of wood-rotting fungi was taken out by an inoculating loop and placed in a bacterial incubator. The pine wood samples were placed in a conical flask containing wood-rotting fungi and stored for 12 weeks (Figure 1).

#### 2.2.4. Specimen Characterization

In this section, to analyze the effect of nano-xylan on the anticorrosive properties of pine wood, the microstructure, physical and mechanical properties, and functional group changes of the samples were tested and characterized with FTIR, SEM, and XRD.

(1)FTIR analysis

FTIR spectroscopy was conducted to examine the functional groups of nano-xylan and pine wood using a Nicolet 6700 FTIR spectrometer (Thermo Fisher Scientific Co., Ltd., Waltham, MA, USA) over the wavenumber range of 3500–4000 cm^−1^ with a scanning rate of 32 scans per min and resolution of 4 cm^−1^.

(2)SEM analysis

SEM examinations were conducted to investigate the morphology of the pine wood after fungal infection. The specimens were examined with a QUANTA 200 SEM (FEI Company, Eindhoven, The Netherlands). A cross section with a thickness lower than 3 mm was selected for SEM observations. The specimens were coated with platinum to improve the surface conductivity and observed at an acceleration voltage of 15 kV.

(3)XRD analysis

Changes in the crystallinity of pine wood specimens before and after modification were analyzed by using a D/max2200 X-ray diffractometer. The scanning range of the sample was 5–40° (2θ) and the scanning speed was 5°/min.

(4)Drug loading, mass loss rate, and mechanical performance degradation

The specimens were put into the vacuum-pressurized impregnation tank and soaked for 50 min under vacuum-pressurized conditions; the impregnated samples were taken out, and the surface preservatives were dried up with sterile blotting paper to obtain the impregnated samples (the average of five values obtained from identical specimens was used as the result).

The drug loading of the specimens was calculated using Equation (1) according to LY/T 1283-2011 “Laboratory Experimental Method for Toxicity of Wood Preservatives to Decaying Bacteria”.
R = (m_2_ − m_1_) × 10C/V(1)

R is the drug loading in the specimen (Kg/m^3^); m_1_ and m_2_ are the weight of specimen before and after immersion, respectively (g); C is the preservative concentration; V is the volume of the treated specimen.

After the infected specimens were removed from the conical flask for wood-rotting, the surface hyphae were taken out. The specimens were then stripped and completely oven-dried at 105 °C. The weight of the dried sample was measured, and the mass loss rate was calculated according to the standard GB/T13942.1-92.

Dried samples (20 mm × 20 mm × 300 mm) were put into an oven at 22 °C and a relative humidity of 65% to adjust their equilibrium moisture content to 12%. Then, the modulus of rupture and modulus of elasticity of the samples were measured according to GB/T 1937-2009 standards, respectively.

## 3. Results and Discussion

### 3.1. Correlation between Additive Loading and Treatment Parameters

The treatment of pine wood can be separated into two main processes: heat modification with high-temperature, high-pressure steam and vacuum impregnation. The influence of vacuum parameters on the additive loading was first investigated using steam heat treatment at a temperature and pressure of 140 °C and 0.4 MPa for 30 min. A vacuum pressure of 0.06 MPa, vacuum time of 30 min, and nano-xylan concentration of 6% were selected as the base impregnation conditions. Then, the control variate method was applied. The ranges for the vacuum degree, vacuum time, and concentration of nano-xylan were [0.02, 0.1] MPa, [10, 50] min, and [2%, 10%], respectively. The influence of these heat modification parameters on the additive loading is demonstrated in Figure 2. The additive loading increased upon raising the vacuum degree, vacuum time, and nano-xylan concentration. As can be seen from Figure 2a–c, the change in nano-xylan concentration had the greatest effect on the increase of sample additive loading, followed by the absolute value of the vacuum degree. Compared with the samples treated with 2% nano-xylan and 10 min vacuum time, respectively, the mass loss rate of the sample treated with 10% nano-xylan increased from 1.06 to 9.75% and 1.01 to 8.23%. Nano-xylan has strong penetration due to its small size; can penetrate wood cell cavity and intercellular space to improve its drug loading; and the higher the concentration of nano-xylan, the more significant the drug loading increase.

The vacuum impregnation method produced a pressure difference between intracellular and extracellular structures inside the wood. Then, air in the cell cavity and pit cavity was easily filled with the nano-xylan solution. In addition, the volume of the wood expanded under the vacuum, which increased the cell spacing and enhanced the impregnation depth.

Correlation analysis was conducted between the heat modification parameters and additive loading. The investigation was also conducted under fixed vacuum impregnation conditions with a nano-xylan concentration of 6%, vacuum degree of −0.06 MPa, and vacuum time of 30 min. Treatment under steam pressure of 0.1 to 0.8 MPa and treatment time of 10 to 50 min were selected as the base impregnation conditions. The influence of the heat modification parameters on the additive loading is demonstrated in Figure 3a,b. Heat treatment enhanced the nano-xylan loading, and the loading rate increased upon raising the steam pressure, temperature, and treatment time. Additionally, the change in stream pressure and tempreature had the greatest effect on the increase of sample drug loading. Compared with the samples treated with 100 °C and 0.1 MPa, the drug loading of the sample treated with 170 °C and 0.8 MPa increased from 7.93 to 11.37%.

Some hemicelluloses, extractives, and resin in pine wood were removed via heat treatment under high temperature, high-pressure steam, which increased the porosity of samples and permeability of the vessels. In addition, the destruction of the thinner pit membrane on the wood cell wall caused by steam promoted the penetration of nano-xylan into the intercellular spaces and cell cavities. More transport channels were generated under a higher steam pressure and temperature, which then lead to a higher additive loading.

### 3.2. Optimization of the Loading Rate Using Response Surface Method

The response surface method was applied to optimize the additive loading rate through heat modification and vacuum impregnation. The four most significant factors were selected for optimization: steam pressure and temperature, heat treatment time, vacuum degree, and vacuum impregnation time. The experimental design scheme and ranges of the independent variables are presented in Table 1. A total of twenty-nine combinations were designed (Table 2), which included five replicates at the center point. Optimization was conducted under a nano-xylan concentration of 10%.

The additive loading based on the design parameters in Table 2 was also summarized in Table 2. A regression equation between the additive loading and the design parameters was built as follows: Y*_drug load_*/% = 11.24 + 0.78*A* + 0.42*B* + 245*C* + 1.42*D*. The model’s correlation of determination *R*^2^ was 0.9462, which indicates a good fit. The high correlations between the actual and predicted values and normal probability, in which all points were close to the same line and the corresponding cumulative percentage point, was almost less than 0.6, as shown in Figure 4a,b, respectively. These indicate that the function of the variable conformed to a normal distribution, and the experimental values were similar to the predicted values obtained for the fitted model, indicating the data were reliable, and the predicted optimum exactly matched the experimental optimum.

Three-dimensional contour plots between the independent variables and additive loading and test of significance for the regression coefficientsare shown in Figure 5 and Table 3. Figure 5a shows the effect of steam pressure, temperature and vacuum, as well as their interactions on the drug loading of the specimen. Among them, the specimen drug loading increases with the increase in steam treatment time, temperature, and vacuum degree, while the model significant value of the interaction between the two models is small, indicating that the interaction between A × C is significant; Figure 5b shows the effect of steam treatment time and vacuum degree and their interaction on the specimen drug loading; it is observed that the specimen drug loading tends to increase with the increase in steam treatment time and vacuum level, while the model significant value of the interaction between them is slightly smaller, indicating that the interaction between B × C is more significant. It can be seen from Figure 5c that the drug loading tends to increase with the increase in vacuum treatment time and vacuum degree, while the significant value of the model of the interaction effect is larger, indicating that the interaction effect between C × D is not significant. In addition, the highest additive loading (14.83%) was achieved at the steam pressure and temperature of 0.8 MPa and 170 °C, heat treatment time of 50 min, vacuum degree of 0.08 MPa, and vacuum impregnation time of 50 min. The optimized additive loading of 14.83% was also in accordance with the predicted value.

Figure 5a shows that the additive loading increased upon increasing the temperature and pressure from 150 °C and 0.5 MPa to 170 °C and 0.8 MPa. Similarly, the additive loading markedly increased with the absolute value of vacuum degree and vacuum time, as shown in Figure 5b. The analysis shows that the vacuum degree had the greatest influence on the additive loading, followed by vacuum time and steam treatment.

### 3.3. Mechanism Analysis Using FTIR, XRD, and SEM

The modification mechanism of the nano-xylan treatment and heat modification on the pine wood was first examined with FTIR spectroscopy. The examined spectra for the original specimen, the nano-xylan-treated specimen, and the specimen treated with high-temperature, high-pressure steam are shown in Figure 6. Heat treatment was conducted at 170 °C and 0.8 MPa for 50 min. Compared with the spectrum of the untreated sample (Curve a), an obvious decrease in the adsorption peak at 3350 cm^−1^ was observed for the wood treated with steam (Curve c). The steam treatment may have released acetic acid from hemicelluloses, which then hydrolytically degraded the hemicelluloses’ chains, resulting in the formation of an ether bond and a decrease in the number of free hydroxyl groups [30]. Furthermore, new adsorption peaks at 830 cm^−1^ and 760 cm^−1^ were observed for the specimen treated with nano-xylan, which was correlated with the β-D-xylopyrose bonds (Curve b). The presence of these two bonds indicated the grafting of nano-xylan to pine wood.

Crystallinity affects wood properties, and the influence of nano-xylan treatment and heat modification on the crystallinity of pine wood was examined with XRD (Figure 7). The treatment with high-temperature, high-pressure steam significantly enhanced the crystallinity of the pine wood. Figure 8 shows that compared with untreated pine wood, the crystallinity of the wood modified with steam significantly improved upon increasing the steam temperature and pressure. The crystallinity of pine wood increased from 47.2% to 56.8% after the heat treatment. The hydroxyl groups between the cellulose molecular chains in the semi-crystalline amorphous region of cellulose combined to form ether bonds. Then, microfibrils in the amorphous region were more orderly arranged and oriented toward the crystalline region, which increased the crystallinity of pine wood.

The morphology of the pine wood after fungal infection was examined, and the results are shown in Figure 8. The cell cavity of the original specimen had a clean internal structure and regular cell wall pit shape, as shown in Figure 8a. After infection by the wood-rot fungi for 12 weeks, a dense distribution of the hyphae clusters was identified inside the cells of the pine wood, as indicated in Figure 8c. An obvious crack was observed along the edges of the pit, and the wood cells were severely damaged. As shown in Figure 8b, the deterioration of the pit was significantly resolved for the fungi-infected specimen after nano-xylan treatment, and the cell walls remained mostly intact. The number of hyphae in the cell cavity was significantly reduced, and the formation of hyphae clusters cannot be observed, showing that nano-xylan treatment enhanced the preservative and antibacterial performance of pine wood.

**Figure 8 materials-16-03976-f008:**
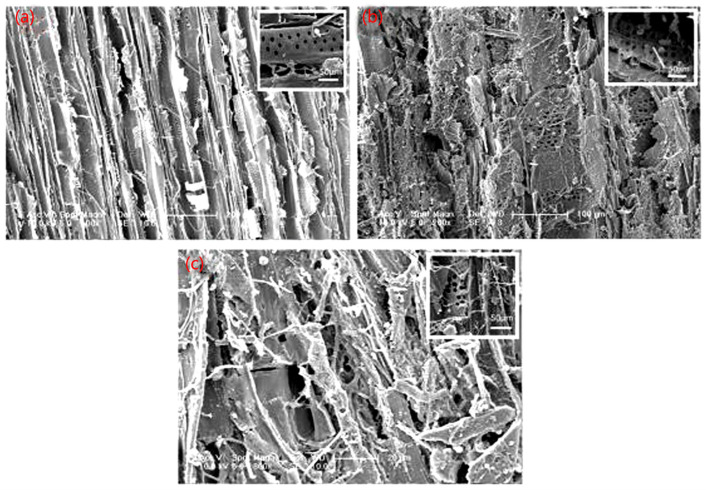
SEM images of the influence of nano-xylan treatment on fungal infection resistance. (**a**) Reference sample; (**b**) infected sample modified with nano-xylan; (**c**) infected sample without treatment.

### 3.4. Analysis of Mass Loss Rate and Degradation of Mechanical Performance

The influence of treatment nano-xylan on the integrity and mechanical performance after infection was examined. The correlation between the measured mass loss rate and the concentration of the nano-xylan is shown in Figure 9. Treatment was conducted under the optimized conditions in Section 3.2. The weight loss rate of the sample decreased upon increasing the nano-xylan concentration. Compared with the untreated sample, the mass loss rate of the sample treated with 10% nano-xylan decreased from 38 to 22%.

The examined strength loss for the pine wood after infection by the fungi is shown in Figure 10. The modulus of rupture and modulus of elasticity of the original samples were 106.37 MPa and 12.31 GPa, respectively. After infection for 12 weeks, the modulus of rupture and modulus of elasticity of the untreated samples decreased to 67.81 MPa and 7.36 GPa, respectively. A smaller strength loss was observed for the samples modified with nano-xylan. When the nano-xylan concentration was 10%, the modulus of rupture and modulus of elasticity of the modified pine wood after a 12-week infection remained at 89.5 MPa and 9.31 GPa, respectively. These results further support that nano-xylan modification enhanced the anti-fungal performance of pine wood.

Cellulose, hemicelluloses, and lignin in the wood are tightly cross-linked to give the wood good mechanical properties. Among them, cellulose plays a load-bearing role in the wood cell walls, and hemicelluloses and lignin increase the cell wall rigidity. White-rot fungi decompose cellulose and hemicelluloses; change the original chemical component ratio of wood; and break the chemical bonds that bind cellulose, hemicelluloses, and lignin, thereby reducing the mechanical properties of wood. Nano-xylan penetrated the cell membrane of bacteria due to its small size, where it can combine with proteins and nucleic acids in the cell and alter the selective permeability of the cell membrane. These effects can disrupt the normal physiological functions of the cells and improve the mechanical properties of the infected samples.

## 4. Conclusions

In this study, the nano-xylan was used as a preservative to improve the the drug loading and fungi resistance (especially against white-rot fungi) of pine wood (*Pinus massoniana Lamb*), determine the best pretreatment, nano-xylan modification process, and analyze the antibacterial mechanism of nano-xylan. The main findings are as follows.

(1)The nano-xylan loading of pine wood was enhanced via modification with high-temperature and high-pressure steam and impregnation under a vacuum. The additive loading generally increased upon increasing the steam pressure and temperature, heat-treatment time, vacuum degree, vacuum time, and nano-xylan concentration. The optimal additive loading (14.83%) was achieved at a steam temperature–pressure of 170 °C−0.8 MPa, heat treatment time of 50 min, vacuum degree of 0.08 MPa, and vacuum impregnation time of 50 min.,(2)High-temperature, high-pressure steam treatment enhanced the crystallinity of pine wood and decreased the number of free hydroxyl groups. Nano-xylan was grafted to the pine wood and inhibited the formation of hyphae clusters and the deterioration of the pine wood microstructure.(3)The weight loss and mechanical performance degradation were mitigated via nano-xylan modification, and the improvement effect increased upon increasing the nano-xylan concentration.

Based on abundant experimental tests and analyses, we see that the adhesion of nano-xylan in wood is not strong and easily produces water loss, which weakens its preservative performance. Meanwhile, the preservative performance of a single nano-xylan is relatively weak. Therefore, the future perspectives and development focus will be carried out from the following aspects:(1)Nano xylan can be compounded with Zn and Cu ions to obtain a new preservative with better anticorrosive properties.(2)Apply nano-xylan and its compound preservative to bamboo, straw, and other biomass materials.

## Figures and Tables

**Figure 1 materials-16-03976-f001:**

Uninfected and infected sample.

**Figure 2 materials-16-03976-f002:**
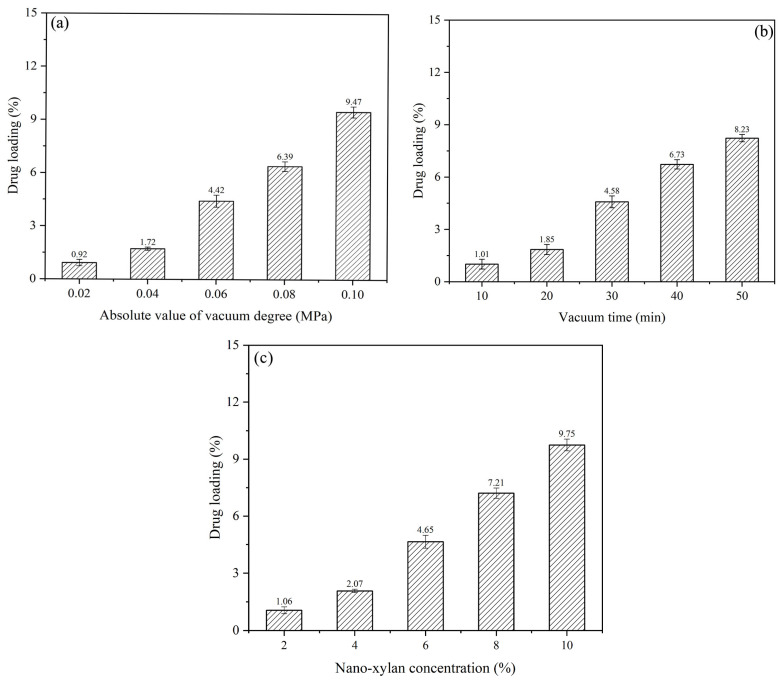
The influence of vacuum impregnation and nano-xylan concentration on the additive loading: (**a**) absolute value of vacuum degree; (**b**) vacuum time; (**c**) nano-xylan concentration.

**Figure 3 materials-16-03976-f003:**
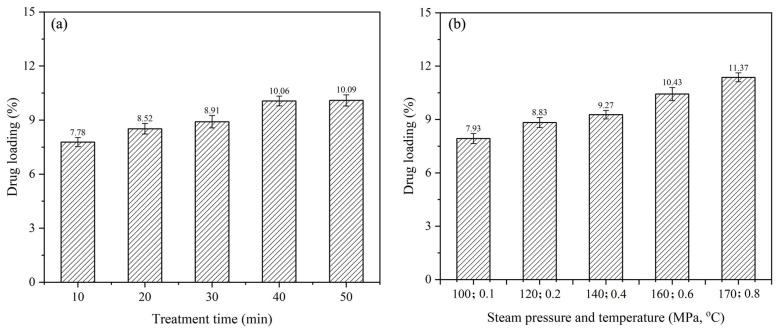
Effect of high-temperature, high-pressure steam treatment on additive loading: (**a**) treatment time; (**b**) steam pressure and temperature.

**Figure 4 materials-16-03976-f004:**
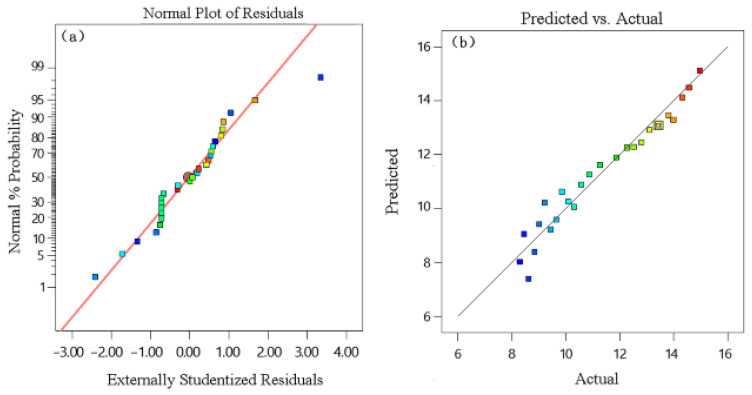
Diagnostics of the models’ equations: (**a**) normal plot of residuals; (**b**) predicted vs. actual.

**Figure 5 materials-16-03976-f005:**
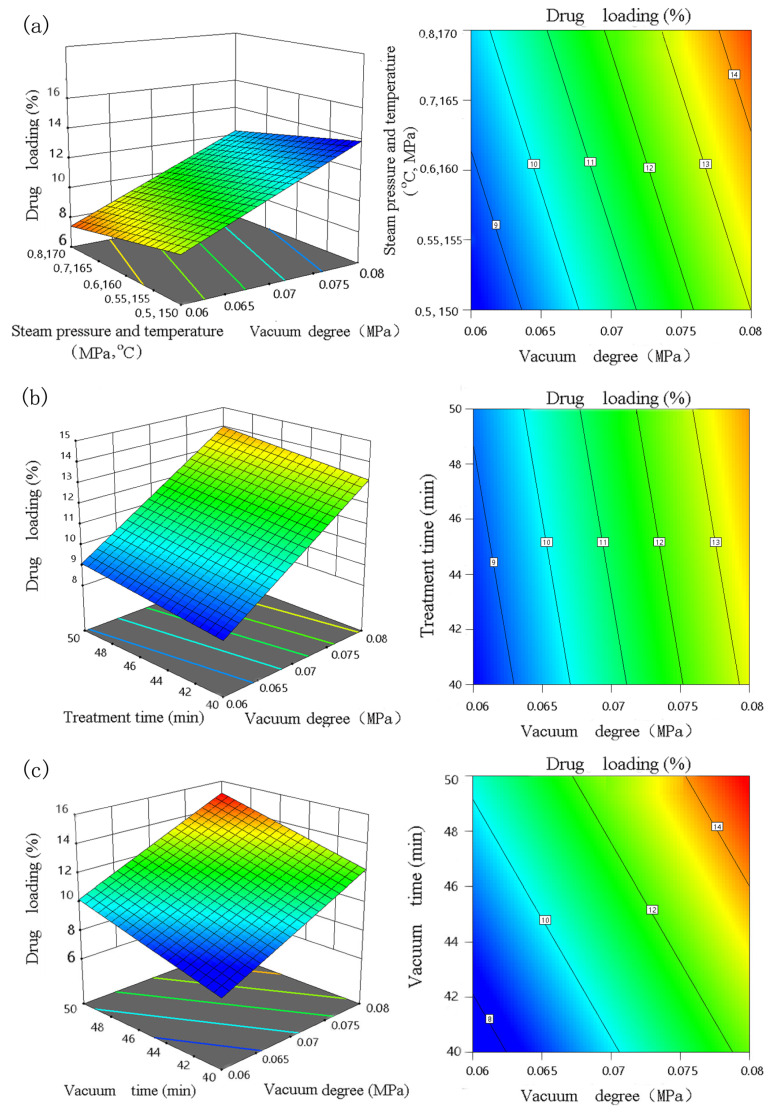
Response surface and contour analysis: (**a**) steam pressure, temperature, and absolute value of vacuum degree; (**b**) treatment time and absolute value of vacuum degree; (**c**) vacuum time and absolute value of vacuum degree.

**Figure 6 materials-16-03976-f006:**
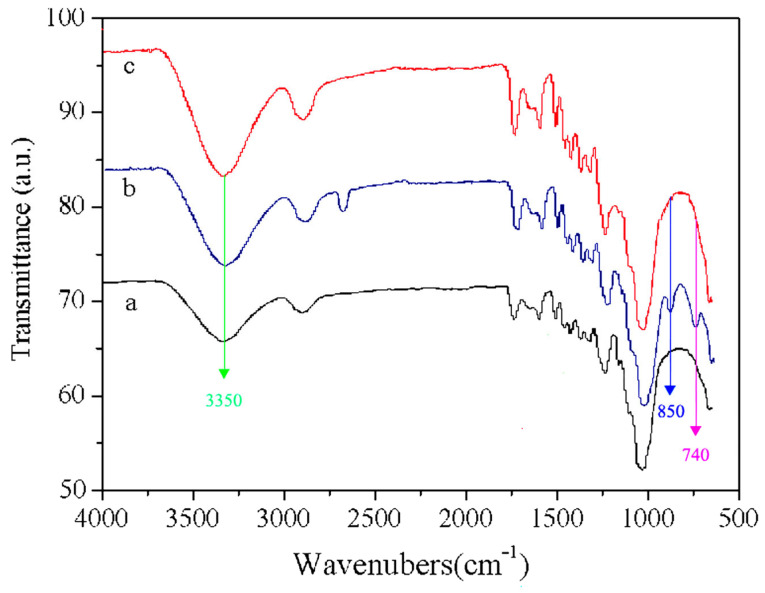
FTIR spectra of the pine wood: (a) reference sample; (b) sample modified with nano-xylan; (c) sample modified with high-temperature, high-pressure steam.

**Figure 7 materials-16-03976-f007:**
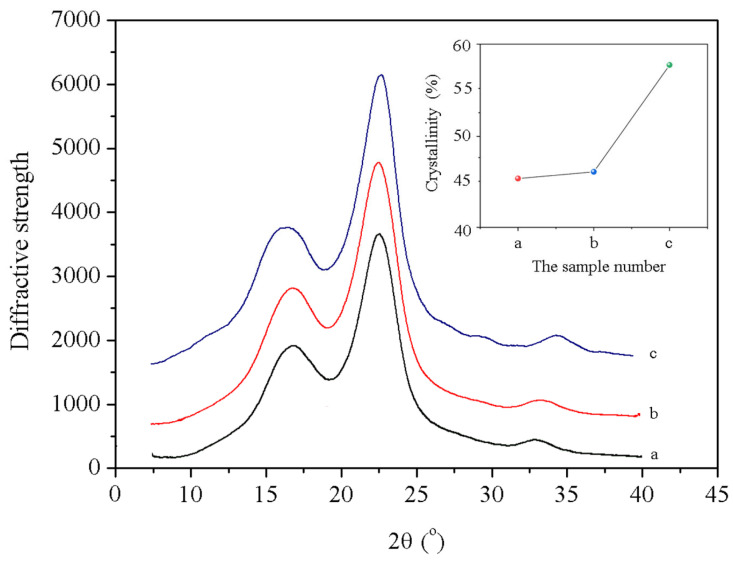
XRD patterns obtained for the modified wood: (a) reference sample; (b) sample modified with nano-xylan; (c) sample modified with high-temperature, high-pressure steam.

**Figure 9 materials-16-03976-f009:**
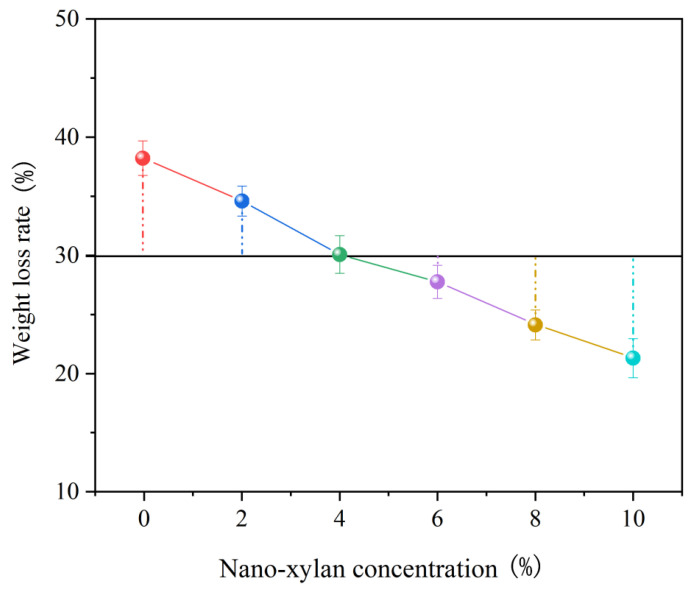
Correlation between the weight loss rate and concentration of nano-xylan.

**Figure 10 materials-16-03976-f010:**
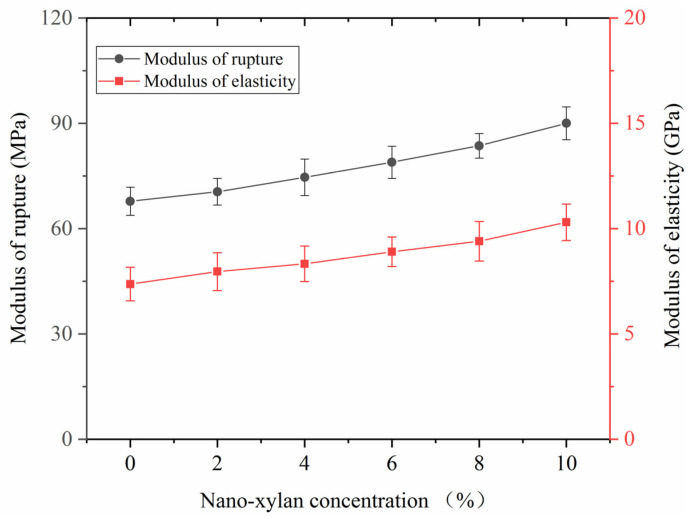
Modulus of rupture and modulus of elasticity analysis of the samples after fungi infection.

**Table 1 materials-16-03976-t001:** Factors and levers of the high-temperature, high-pressure steam pretreatment-assisted vacuum impregnation method used in response surface design.

Factor	Code	Levels		
Steam pressure (MPa) and Temperature (°C)	A	0.5, 150	0.6, 160	0.8, 170
Treatment time (min)	B	40	45	50
Absolute value of vacuum degree (MPa)	C	−0.06	−0.07	−0.08
Vacuum time (min)	D	40	45	50

**Table 2 materials-16-03976-t002:** Independent variables of the central composite design and the results of response surface analysis.

Run	A	B	C	D	Additive Loading (%)
1	150	45	0.07	50	11.89
2	160	45	0.07	45	10.89
3	160	45	0.07	45	10.89
4	160	50	0.07	50	13.44
5	160	45	0.08	40	12.53
6	160	40	0.08	45	14
7	150	45	0.06	45	8.31
8	160	40	0.07	50	12.28
9	150	45	0.07	40	8.46
10	170	45	0.06	45	9.66
11	160	45	0.06	40	8.63
12	170	45	0.07	40	9.87
13	160	45	0.07	45	10.89
14	160	50	0.08	45	13.33
15	160	50	0.07	40	10.11
16	150	40	0.07	45	10.32
17	160	40	0.06	45	8.85
18	160	45	0.06	50	9.23
19	150	45	0.08	45	13.11
20	160	45	0.07	45	10.89
21	170	45	0.08	45	14.28
22	170	45	0.07	50	13.83
23	160	40	0.07	40	9.03
24	160	45	0.07	45	10.89
25	160	45	0.08	50	14.07
26	170	50	0.07	45	12.82
27	160	50	0.06	45	9.45
28	170	40	0.07	45	11.27
29	150	50	0.07	45	10.58

**Table 3 materials-16-03976-t003:** Test of significance for the regression coefficients.

Source	Sum of Squares	dF	Mean Square	F-Value	*p*-Value
Model	105.05	14	26.26	98.42	<0.0001
A	7.30	1	7.30	27.36	<0.0001
B	1.66	1	1.66	6.21	0.0200
C	71.98	1	71.98	269.76	<0.0001
D	24.11	1	24.11	90.36	<0.0001
AB	21.56	1	31.2	24.81	<0.0001
AC	11.28	1	10.07	18.53	<0.0001
AD	1.35	1	2.23	27.16	<0.0112
BC	0.26	1	0.48	31.85	<0.0029
BD	0.06	1	0.67	13.07	<0.0546
CD	3.98	1	14.86	8.55	<0.0731
A^2^	98.32	1	31.57	275.14	<0.0001
B^2^	106.37	1	72.96	571.66	<0.0001
C^2^	198.67	1	168.22	876.21	<0.0001
D^2^	85.34	1	54.97	635.17	<0.0001
Residual	6.40	24	0.2668	/	
Lack-of-fit	6.40	20	0.3202	3.85	0.0021
Pure error	0.0000	4	0.0000	/	/
Cotal deviation	111.46	48	/	/	/

## Data Availability

Not applicable.

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
