# Peer review of "Enhanced Preservative Performance of Pine Wood through Nano-Xylan Treatment Assisted by High-Temperature Steam and Vacuum Impregnation"

_materials, 2023, doi:10.3390/ma16113976_

Round 1

Reviewer 1 Report (Previous Reviewer 1)

The authors submitted a work about the enhancement of the preservative performance of Pine wood. The quality of the work is acceptable, but the following major points need to be revised:

1. The introduction section can be improved. Please, refer to the following recent studies carried out on Pine wood and that can surely enrich the literature background: 10.1016/j.finmec.2022.100094

2. Section 2.1 is a mere list of materials. Please make it more descriptive and fluid to read. In addition, it could be nice to see a picture of all the used materials.

3. Sections 2.2, 2.3 and 2.4 are too short and could not be considered proper Sections. They should be joined in a single Section and described properly, maybe through a bullet list. It could be useful for the readers if the authors could add schematic of the process (like a flowchart, for example). In addition, please note that it is not recommendable to start sentenced with numbers (e.g. “50g of corncob…”).

4. Figure 1 and Figure 2 are only cited in the text. The different subplots (a, b, c …) need to be explained in the text, not only in the figure caption.

5. In general, none of the Figures is described properly. Both in the text and in the figure captions, all the figures’ elements (curves, markers, error bars etc.) and subplots must be described in order to be clearly and quickly understandable by the reader. In addition, Figures need to be discussed more deeply; at the moment, they are just presented without being described and commented satisfactorily. This makes the readability of the work tough and not fluid. The authors are strongly invited to fix the above issues.

6. In the conclusions Section the authors should discuss possible future perspectives and developments of their work.

Dear authors, the English must be improved. In particular, some Sections are hard to read because they are mere list of actions. Language and writing should be more fluid for helping the readabilidy.

Author Response

1. The introduction section can be improved. Please, refer to the following recent studies carried out on Pine wood and that can surely enrich the literature background: 10.1016/j.finmec.2022.100094

Response: Thanks for your question. We have added the literature to enrich the literature background in the revised manuscript.

2. Section 2.1 is a mere list of materials. Please make it more descriptive and fluid to read. In addition, it could be nice to see a picture of all the used materials.

Response: Thanks for your question. We have made the section 2.1 more descriptive and fluid.

3. Sections 2.2,2.3 and 2.4 are too short and could not be considered proper Sections. They should be joined in a single Section and described properly, maybe through a bullet list. It could be useful for the readers if the authors could add schematic of the process (like a flowchart, for example). In addition, please note that it is not recommendable to start sentenced with numbers (e.g. “50g of corncob…”).

Response: Thanks for your question. We have revised the problem of the ‘start sentenced with numbers’ and the titles of 2.2,2.3 and 2.4 in the Experimental section. 

4. Figure 1 and Figure 2 are only cited in the text. The different subplots (a, b, c …) need to be explained in the text, not only in the figure caption.

Response: Thanks for your question. We have revised it in the 3.1 section.

5. In general, none of the Figures is described properly. Both in the text and in the figure captions, all the figures’ elements (curves, markers, error bars etc.) and subplots must be described in order to be clearly and quickly understandable by the reader. In addition, Figures need to be discussed more deeply; at the moment, they are just presented without being described and commented satisfactorily. This makes the readability of the work tough and not fluid. The authors are strongly invited to fix the above issues.

Response: Thanks for your question. We have discussed more deeply on the content of Figures in the revised manuscript.

6. In the conclusions Section the authors should discuss possible future perspectives and developments of their work.

Response: Thanks for your question. We have discussed possible future perspectives and developments of our work in the conclusions section.

7. the English must be improved. In particular, some Sections are hard to read because they are mere list of actions. Language and writing should be more fluid for helping the readabilidy.

Response: We have called upon professional editing services to enhance the language quality of our paper.

Reviewer 2 Report (Previous Reviewer 2)

Accept in present form

Author Response

Thank you for your comments

Reviewer 3 Report (New Reviewer)

The publication submitted for review is interesting. It deals with wood protection by pressure impregnation and nano-xylene treatment. The publication is well written. The structure of the work is correct. The methodology is well chosen. The figures are also well presented, and the data in the tables are clearly given.

Minor comments:

- the purpose of the work is not clear from the introduction,

- the authors should precisely define the purpose of the work,

- descriptions of the work performed should be given in the methodology of the work,

- descriptions of studies from the discussion of results should be moved to the methodology or removed if they are repetitive,

- give the Latin name of the wood used,

- if the Authors connect the dots in Fig. 1, it is advisable to provide a trend line. If the authors are in doubt about what is in the discontinuity points, it may be better to make bar charts,

- the paper lacks an extended statistical analysis to indicate the significant effect of the influence of a specific factor,

- what sample dimensions were used to analyse the mechanical characteristics?

Best Regards,

no other comments

Author Response

1. the purpose of the work is not clear from the introduction,

Response: Thanks for your question. We have precisely defined the purpose of the work in the introduction section.

2. the authors should precisely define the purpose of the work,

Response: Thanks for your question. We have precisely defined the purpose of the work in the revised manuscript.

3. descriptions of the work performed should be given in the methodology of the work,

Response: Thanks for your question. We have given the descriptions of the work performed in the revised manuscript.

4. descriptions of studies from the discussion of results should be moved to the methodology or removed if they are repetitive,

Response: Thanks for your question. We have removed the descriptions of studies in the revised manuscript.

5. give the Latin name of the wood used

Response: Thanks for your question. We have specified the scientific name of the pine wood in the revised manuscript.

6. if the Authors connect the dots in Fig. 1, it is advisable to provide a trend line. If the authors are in doubt about what is in the discontinuity points, it may be better to make bar charts,

Response: Thanks for your question. We have revised the Fig. 1 and Fig. 2 in the revised manuscript.

7. the paper lacks an extended statistical analysis to indicate the significant effect of the influence of a specific factor,

Response: Thanks for your question. We have added the statistical analysis to indicate the significant effect of the influence of a specific factor in the revised manuscript.

8. what sample dimensions were used to analyse the mechanical characteristics?

Response: Thanks for your question. We have specified the specimen's size for the static bending test in the last paragraph of the Experimental section.

Reviewer 4 Report (New Reviewer)

The aim of the study was to enhance the properties of pine wood, specifically its durability against white rot fungi, through treatment with nano-xylan. While I am in favor of publishing this manuscript in the Materials journal, I would like to suggest that the authors consider the following comments to improve the manuscript further.

1- In the Abstract and Conclusions, specify that the modification process enhances the pine wood against white rot fungi specifically, as there are other main groups of fungi that may behave differently towards the modified wood.

2- Rewrite the aim included in the Abstract to clearly indicate the purpose of the study.

3- Please specify the specimen's size for the static bending test.

4- Page 3, Lines 94-96: Summarize this paragraph and state the aims of the study directly instead of referring to sections 2 and 3.

5- The authors should specify the scientific name of the pine wood used in the study.

6- The authors should explain how they calculated the retention (drug loading).

7- Table 1: Provide a suitable indicative title and ensure that the units are presented uniformly within brackets.

8- Figure 3 (b): Include the correlation equation or the correlation coefficient (r) in the graph.

9- Line 228, P. 8: Correction - R2 is the coefficient of determination, not the correlation coefficient.

The manuscript contains several grammatical errors. It is recommended that the authors thoroughly proofread and edit the entire manuscript to ensure its grammatical accuracy.

Author Response

1. In the Abstract and Conclusions, specify that the modification process enhances the pine wood against white rot fungi specifically, as there are other main groups of fungi that may behave differently towards the modified wood.

Response: Thanks for your question. We have specified the modification process enhances the pine wood against white rot fungi specifically in the Abstract and Conclusion section.

2. Rewrite the aim included in the Abstract to clearly indicate the purpose of the study.

Response: Thanks for your question. We have rewritten the aim included in the Abstract to clearly indicate the purpose of the study.  

3. Please specify the specimen's size for the static bending test.

Response: Thanks for your question. We have specified the specimen's size for the static bending test in the last paragraph of the Experimental section.

4. Page 3, Lines 94-96: Summarize this paragraph and state the aims of the study directly instead of referring to sections 2 and 3.

Response: Thanks for your question. We have summarized this paragraph and state the aims of the study directly instead of referring to sections 2 and 3 in the last paragraph of the introduction section.

5. The authors should specify the scientific name of the pine wood used in the study.

Response: Thanks for your question. We have specified the scientific name of the pine wood in the revised manuscript.

6. The authors should explain how they calculated the retention (drug loading).

Response: Thanks for your question. We have explained how to calculated the drug loading in the 2.2 section.

7. Table 1: Provide a suitable indicative title and ensure that the units are presented uniformly within brackets.

Response: Thanks for your question. We have revised the indicative title and ensured the units are presented uniformly within brackets.

8. Figure 3 (b): Include the correlation equation or the correlation coefficient (r) in the graph.

Response: Thanks for your question. I am very sorry. In this paper, the data processing and drawing of normal plot of residuals and predicted vs. actual were all completed by Design Expert software. Unfortunately, the correlation equations or coefficients in the graph could not be calculated using this software.

9. Line 228, P. 8: Correction - R2 is the coefficient of determination, not the correlation coefficient.

Response: Sorry for my negligence, we have revised it according to your suggestion.

10. The manuscript contains grammatical errors. It is recommended that the authors thoroughly proofread and edit the entire manuscript to ensure its grammatical accuracy.

Response: We have called upon professional editing services to enhance the language quality of our paper.

Round 2

Reviewer 1 Report (Previous Reviewer 1)

The paper can be accepted for publication

Author Response

Thank you for your comments

This manuscript is a resubmission of an earlier submission. The following is a list of the peer review reports and author responses from that submission.

Round 1

Reviewer 1 Report

The authors submitted a work about the enhancement of the antiseptic performance of Pine wood. The quality of the work is acceptable, but the following major points need to be revised:

1. The introduction section needs to be extended. In the present form it does not provide a clear frame in which the manuscript will be placed. Please, add a couple of additional references. For example, it could be interesting to highlight that thermal treatments, despite having beneficial effects on the drug loading, may trigger thermo-mechanical damage, due to the hygroscopic nature of wood. Please, refer to the following recent studies carried out on Pine wood and that can surely enrich the literature background: o 10.1016/j.finmec.2022.100094 o 10.3390/atmos13050791

2. At the end of the introduction section, three acronyms are cited (SEM, FTIR and XDR) but they are not explained. When acronyms appear for the first time in a manuscript, they must first be written in the extended version.

3. At the end of the introduction section, in general, there is a short summary about the manuscript’s structure. Please add it. (For example: “Section 2 deals with…, focusing on… while Section 3 is about… etc etc”)

4. Section 2.1 is a simple list of materials. Please make it more fluid to read and, if possible, add a figure showing all the used materials.

5. Section 2.2 is a list of steps. Before describing the actual steps of the treatment, a couple of lines should be added for introducing the new section. For instance: “In this section, the antiseptic treatment of the Pine wood with the synthesized nano-xylan is explained. It has been carried out in two main steps that are described as follows…”.

6. In Figure 1 please add the labels a) and b) to the two images. In addition, please explain better and with more details in the text what Figure 1 represents.

7. Figure 2 (and all its images) needs to be explained better and with more details in the text

8. Section 2.4 is a list of steps. As said for Section 2.2, please add a couple of lines for introducing the section.

9. In general, none of the Figures is described properly. Both in the text and in the figure captions, all the figures’ elements (curves, markers, error bars etc.) and subplots must be described in order to be clearly and quickly understandable by the reader. In addition, Figures need to be discussed more deeply; at the moment, they are just presented without being described and commented satisfactorily. This makes the readability of the work tough and not fluid. The authors are strongly invited to fix the above issues.

10. Finally, in the conclusions section, it is recommended to stress the steps that may be carried out in the near future and the perspectives expected by the authors.

Author Response

Point 1: The introduction section needs to be extended. In the present form it does not provide a clear frame in which the manuscript will be placed. Please, add a couple of additional references. For example, it could be interesting to highlight that thermal treatments, despite having beneficial effects on the drug loading, may trigger thermo-mechanical damage, due to the hygroscopic nature of wood. Please, refer to the following recent studies carried out on Pine wood and that can surely enrich the literature background: o 10.1016/j.finmec.2022.100094 o 10.3390/atmos13050791

Response 1: Thanks for your question. We have added a couple of additional references in the introduction section.

Point 2: At the end of the introduction section, three acronyms are cited (SEM, FTIR and XDR) but they are not explained. When acronyms appear for the first time in a manuscript, they must first be written in the extended version.

Response 2: Thanks for your question. The acronyms have been written in the extended version at the end of the introduction section.

Point 3: At the end of the introduction section, in general, there is a short summary about the manuscript’s structure. Please add it. (For example: “Section 2 deals with…, focusing on… while Section 3 is about… etc etc”)

Response 3: Thanks for your question. We have added a short summary about the manuscript’s structure at the end of the introduction section

Point 4: Section 2.1 is a simple list of materials. Please make it more fluid to read and, if possible, add a figure showing all the used materials.

Response 4: We tried to make a figure showing all the used materials, but it looked very ugly in the text, so we gave up this idea

Point 5: Section 2.2 is a list of steps. Before describing the actual steps of the treatment, a couple of lines should be added for introducing the new section. For instance: “In this section, the antiseptic treatment of the Pine wood with the synthesized nano-xylan is explained. It has been carried out in two main steps that are described as follows…”.

Response 5: Thanks for your suggestion. We have added an introducing for Section 2.2.

Point 6: In Figure 1 please add the labels a) and b) to the two images. In addition, please explain better and with more details in the text what Figure 1 represents.

Response 6: Other reviewers considered Figure 1 redundant and suggest to remove it. We have deleted Figure 1 and added more details in the text of what Figure 1 represents.

Point 7: Figure 2 (and all its images) needs to be explained better and with more details in the text.

Response 7: Other reviewers consider Figure 2 (and all its images) is redundant and suggest to remove it. We have deleted Figure 1 and added more details in the text what Figure 1 represents.

Point 8: Section 2.4 is a list of steps. As said for Section 2.2, please add a couple of lines for introducing the section.

Response 8: We have added an introducing for Section 2.4.

Point 9: In general, none of the Figures is described properly. Both in the text and in the figure captions, all the figures’ elements (curves, markers, error bars etc.) and subplots must be described in order to be clearly and quickly understandable by the reader. In addition, Figures need to be discussed more deeply; at the moment, they are just presented without being described and commented satisfactorily. This makes the readability of the work tough and not fluid. The authors are strongly invited to fix the above issues.

Response 9: We have revised in more detail in the description of the figures. 

Point 10: Finally, in the conclusions section, it is recommended to stress the steps that may be carried out in the near future and the perspectives expected by the authors.

Response 10: Thanks for your suggestion. We have added future research needs in the conclusions section.

Reviewer 2 Report

This paper shows interesting data, but the content and results are not enough for original article. It is better to organize the contents significantly by adding detailed explanations of the experimental methods and results, or resubmit.

Author Response

This paper shows interesting data, but the content and results are not enough for original article. It is better to organize the contents significantly by adding detailed explanations of the experimental methods and results, or resubmit.

Response : We have revised in more detail in the abstract, introduction, conclusion and discussion, and conclusion. And we have called upon professional editing services to enhance the language quality of our parer.

Reviewer 3 Report

Dear Authors,

Please see the comments.

First line of the abstract, what do you mean (he), you mean (the).

In the abstract, the degradation on ….., please change on to (of).

Paragraph 2 of introduction, it is found the single sewage…. Please change it to it is found that a single ……

Paragraph 3 of introduction, please be careful, xylan possessed excellent …. Change to possesses and also add (have) instead of (own). 

Paragraph 4 of introduction, please add (s) to the deteriorate

Revise the English of the manuscript.

Section 2.1. raw material, please add the following information (the samples were from sap wood or heart wood, as well as the age of the woods)

Below the figure 1, when the samples placed into the nano-xylan and impregnated in the vacuum chamber, the samples were placed in an oven in order to be dried. Nonetheless, the temperature and the time did not express in the manuscript. Please add.

Section 2.3 Inoculation and cultivation, change took to taken.

In section 2.4. (4), Please indicate the number of repetitions of the experiments.

In section 3.3, with increasing the temperature of heating, the degradation of hemicellulose is also increased, which is indicated in the mentioned part. But with increasing the degradation of hemicellulose, the mechanical properties of woods are also decreased, which is presented in your manuscript. Nonetheless, in the section of XRD, the results were different?

In figure 9, Where are the results of contaminated samples from wood treated with high temperature and high pressure? Did you do this test?

In figure 10 and 11, the results showed that with increasing the amount of nano-xylan concentration, the weight loss of the samples was declined. In figure 11, but why the mechanical properties improved with the addition of more nano-xylan concentration, in spite of the fact that weight loss was declined. 

Author Response

Point 1: Moderate English changes required

Response 1: We have called upon professional editing services to enhance the language quality of our paper.

Point 2: First line of the abstract, what do you mean (he), you mean (the).

Response 2: Sorry for my negligence, we have revised it.

Point 3: In the abstract, the degradation on ….., please change on to (of).

Response 3: Sorry for my negligence, we have revised it.

Point 4: Paragraph 2 of introduction, it is found the single sewage…. Please change it to it is found that a single ……

Response 4: Sorry for my negligence, we have revised it.

Point 5: Paragraph 3 of introduction, please be careful, xylan possessed excellent …. Change to possesses and also add (have) instead of (own). 

Response 5: Sorry for my negligence, we have revised it.

Point 6: Paragraph 4 of introduction, please add (s) to the deteriorate

Response 6: Sorry for my negligence, we have revised it.

Revise the English of the manuscript.

Point 7: Section 2.1. raw material, please add the following information (the samples were from sap wood or heart wood, as well as the age of the woods)

Response 7: Thanks for your question. We have added the information of the pine wood in Section 2.1.

Point 8: Below the figure 1, when the samples placed into the nano-xylan and impregnated in the vacuum chamber, the samples were placed in an oven in order to be dried. Nonetheless, the temperature and the time did not express in the manuscript. Please add.

Response 8: Thanks for your question. We have added the temperature and the time in the manuscript.

Point 9: Section 2.3 Inoculation and cultivation, change took to taken.

Response 9: Thanks for your question. We have changed took to taken.

Point 10: In section 2.4. (4), Please indicate the number of repetitions of the experiments.

Response 10: We did five repeated experiments

Point 11: In section 3.3, with increasing the temperature of heating, the degradation of hemicellulose is also increased, which is indicated in the mentioned part. But with increasing the degradation of hemicellulose, the mechanical properties of woods are also decreased, which is presented in your manuscript. Nonetheless, in the section of XRD, the results were different?

Response 11: For different species of wood, when not treated, the greater the crystallinity, the higher the mechanical properties, but the wood after high-temperature treatment, hemicellulose degradation will also lead to increased crystallinity, but the degradation of hemicellulose, resulting in brittle wood, mechanical properties reduced.

Point 12: In figure 9, Where are the results of contaminated samples from wood treated with high temperature and high pressure? Did you do this test?

Response 12: Sorry for my mistake, we did not do the experiment of high temperature and high pressure treatment of samples. But in the future research, we will use the nano-xylan compounded with Zn+ and Cu+ to improve the wood anti-corrosion. And we will analyze the microstructure of the samples treated with high temperature and high pressure.

Point 13: In figure 10 and 11, the results showed that with increasing the amount of nano-xylan concentration, the weight loss of the samples was declined. In figure 11, but why the mechanical properties improved with the addition of more nano-xylan concentration, in spite of the fact that weight loss was declined. 

Response 13: The mechanical properties were tested after the sample was infected with wood rot fungus for 12 weeks. The mechanical properties of untreated samples decreased the most. The mechanical properties of samples treated with nano xylan decreased slightly. Therefore, compared with untreated samples, the mechanical properties of samples treated with 10% nano-xylan were improved.

Reviewer 4 Report

The manuscript entitled “Enhancement on the Antiseptic Performance of Pine Wood through the Nanoxylan Treatment Assisted by High Temperature Steam and Vacuum Impregnation” describes the effect of combined treatment on wood resistance against fungi. Unfortunately, although potentially interesting, the manuscript in this form is not ready for being published.

Although theoretically aimed at the antifungal effects of the treatment, in fact, the paper is focused on the method to enhance the xylane weight percent gain in wood during the treatment – I would suggest changing the focus of the article or changing the title and focus solely on the optimisation of the method.

The manuscript has several flaws starting from the Introduction part. There are several misstatements and inaccuracies, as well as missed information necessary for the study (e.g. the effect of applied thermal treatment on the wood properties, including permeability and resistance to decay). This theoretical part must be improved, and all the crucial information must be included in discussing the results obtained. I stopped reviewing at some point because the article, experiments and discussion seemed useless without a robust scientific basis. Moreover, the manuscript needs thorough linguistic corrections. Therefore, I recommend the rejection of the paper and the re-submission of an improved article.

Some more detailed comments, questions and suggestions on how to improve the paper can be found in the attached pdf file.

Author Response

Point 1: Extensive editing of English language and style required.

Response 1: We have called upon professional editing services to enhance the language quality of our paper.

Point 2: In the title. I believe this word “antiseptic” is incorrect in this context: "An antiseptic (from Greek á¼€ντί anti, "against"[1] and σηπτικÏŒς sÄ“ptikos, "putrefactive"[2]) is an antimicrobial substance or compound that is applied to living tissue/skin to reduce the possibility of infection, sepsis, or putrefaction. Antiseptics are generally distinguished from antibiotics by the latter's ability to safely destroy bacteria within the body, and from disinfectants, which destroy microorganisms found on non-living objects" [Wikipedia]

Please correct it.

Response 2: Thanks for your suggestion. We have changed “antiseptic” to “preservative”.

Point 3: In the abstract. There is no information about anti-fungal properties of the treated wood. please add some numbers to show how wood resistnce was improved compared to untreated wood.

Response 3: Thanks for your suggestion. We have added some information about anti-fungal properties of the treated wood in abstract section.

Point 4: Paragraph 1 of introduction. What about all the studies on using natural substances for wood treatment against fungal degradation? Please supplement the information, since exensive research has been conducted in this area.

Response 4: Thanks for your suggestion. We have added the supplement information about natural substances for wood treatment against fungal degradation in Paragraph 1 of the introduction.

Point 5: Paragraph 3 of introduction. First of all, xylan is one of the structural polymers of plant cell wall, one of the hemicelluloses - xylans constitute 25–35% of the dry biomass of woody tissues of dicots and lignified tissues of monocots and occur up to 50% in some tissues of cereal grains. This information should be added to the manuscript. Also, please provide more references showing that xylan is a preservative.

Response 5: Thanks for your suggestion. We have added this information to the the manuscript and provided more references showing that xylan is a preservative.

Point 6: Paragraph 3 of introduction. But the provided literature refers to a xylan-chitosan conugate, not to xylan itself - please correct the statement.

Response 6: Thanks for your suggestion. We have corrected the statement.

Point 7: Paragraph 4 of introduction. Please provide some references showing that pine wood has poor permeability. In fact, it is much more permeable than, for example, fir: https://www.nzffa.org.nz/farm-forestry-model/resource-centre/tree-grower-articles/august-2007/responding-to-moisture-how-do-douglas-fir-and-radiata-compare/. Please check the information provided and correct the text.

Response 7: We have provided some references showing that pine wood has poor permeability.

Point 8: Paragraph 4 of introduction. How is this information related to the study? please explain.

Response 8: This paper mainly studies the preparation of nano-xylan and uses it as a preservative to impregnate pine wood. However, the permeability of pine is poor. In order to improve the impregnation efficiency of nano-xylan, it is necessary to pretreat pine with high temperature and high pressure steam to improve the permeability of pine.

Point 9: Is this equipment somehow unique and its structure/mode of action is important for the study? If not, the figure1 is redundant and should be removed.

This figure is redundant. Please remove it.

Response 9: We have deleted Figure 1 and added more details in the text of what Figure 1 represents.